# Lower Limb Deformity in Different Types of Rickets

**DOI:** 10.3390/jcm14238586

**Published:** 2025-12-04

**Authors:** Anahita Mayr, Benjamin Kraler, Catharina Chiari, Gabriele Hartmann, Adalbert Raimann, Gabriel T. Mindler

**Affiliations:** 1Department of Pediatrics and Adolescent Medicine, Division of Pediatric Pulmonology, Allergology and Endocrinology, Medical University of Vienna, Währinger Gürtel 18–20, 1090 Vienna, Austria; 2Vienna Bone and Growth Center, Währinger Gürtel 18–20, 1090 Vienna, Austriagabriel.mindler@oss.at (G.T.M.); 3Department of Pediatric Orthopedics, Orthopedic Hospital Speising, Speisinger Strasse 109, 1130 Vienna, Austria

**Keywords:** rickets, lower limb deformity, hypophosphatemic rickets, calcipenic rickets, XLH, pediatric bone disease

## Abstract

**Background/Objectives:** Rickets is a pediatric disorder caused by impaired mineralization of the growth plate, primarily due to deficiencies in vitamin D, calcium, or phosphate. Lower limb deformities are among the most clinically relevant skeletal manifestations, significantly affecting quality of life. This review aims to summarize and compare lower limb deformities associated with different etiologies of rickets, specifically hypophosphatemic (HPR) and calcipenic (CR) forms. **Methods:** A systematic PubMed search was performed according to PRISMA (Preferred Reporting Items for Systematic Reviews and Meta-Analyses) 2020 guidelines. Of 2056 screened records, 126 studies including 21,568 patients met the inclusion criteria. Data on study characteristics, deformity types, and diagnostic methods were extracted and analyzed descriptively, with subgroup comparison between HPR and CR. **Results:** Among 2924 patients with reported with deformities (1537 CR, 1387 HPR), genu varum and genu valgum predominated, while windswept deformities were less frequent. Sagittal and torsional deformities, particularly femoral and tibial maltorsion or procurvatum, were mainly described in HPR but rarely reported in CR. Only a minority of studies met predefined quality standards regarding radiological assessment and deformity definition. **Conclusions:** Lower limb deformities are prevalent in both HPR and CR but differ significantly in type and documentation quality. While coronal plane deformities are common across both types, axial and sagittal deformities appear under-reported in rickets, particularly in CR. The results highlight the need for standardized diagnostic criteria and improved reporting in order to enhance comparability and clinical management of rickets-related deformities.

## 1. Introduction

Rickets remains a major pediatric health concern worldwide, particularly in low- and middle-income countries where nutritional deficiencies, limited sun exposure, and inadequate access to preventive healthcare contribute to its persistence [1]. The World Health Organization estimates that millions of children are affected globally, underscoring its relevance as a public health priority. In high-income countries, rickets is less common and is predominantly associated with rare genetic or metabolic causes that often require specialized, multidisciplinary care [2].

Pathophysiologically, rickets is defined by defective mineralization of growth plate cartilage, leading to impaired endochondral ossification. It is broadly classified into calcipenic rickets (CR), caused by insufficient calcium availability in the growth plate, and hypophosphatemic rickets (HPR), resulting from chronic hypophosphatemia [3] (Figure 1).

CR most frequently arises from nutritional vitamin D deficiency, although impaired vitamin D activation (e.g., vitamin D-dependent rickets type 1A (VDDR1A)) or end-organ resistance (e.g., vitamin D-dependent rickets type 2A (VDDR2A)) are less common causes [4]. Hypocalcemia in CR triggers secondary hyperparathyroidism, further increasing phosphate losses and exacerbating defective mineralization [5]. CR remains the predominant form worldwide and is strongly linked to poor nutrition, inadequate sun exposure, and malabsorptive disorders [2]. HPR encompasses a heterogeneous group of disorders characterized by renal phosphate wasting. This renal loss is either caused by tubular dysfunction, or by excess fibroblast growth factor 23 (FGF23) activity most commonly due to genetic causes. To date, mutations in at least ten distinct genes have been identified as contributors to these conditions [6,7]. The most frequent subtype is X-linked hypophosphatemia (XLH), caused by pathogenic variants in the *PHEX* gene, which elevate circulating FGF23 and suppress renal phosphate reabsorption and active vitamin D synthesis [8]. Other hereditary forms include autosomal dominant hypophosphatemic rickets (ADHR) and autosomal recessive hypophosphatemic rickets (ARHR) variants involving genes such as *DMP1* and *ENPP1*, while acquired causes such as tumor-induced osteomalacia (TIO) are extremely rare in children but a relevant differential diagnosis in adolescents. Another autosomal recessive form of hypophosphatemic rickets caused by *SLC34A3* mutations, is hereditary hypophosphatemic rickets with hypercalciuria (HHRH) distinguished by hypercalciuria and response to phosphate alone [9].

Skeletal deformities, particularly of the lower limbs, are hallmark manifestations of rickets and sometimes the first signs prompting clinical evaluation. Coronal plane deformities such as genu varum and genu valgum are well recognized, but recent imaging and gait analysis studies—especially in XLH—highlight more complex multiplanar deformities involving the sagittal (e.g., femoral or tibial procurvatum) and axial planes (e.g., femoral torsion, tibial torsion), as well as pelvic malalignment [10,11,12,13,14]. Clinical cases may present with asymmetric or combined deformities, as illustrated in Figure 2. These deformities can contribute to pathological gait patterns, recurrent falls, pain, early-onset joint degeneration, and impaired quality of life [15,16,17].

Standardized orthopedic assessment is essential to characterize these deformities accurately. Long-leg standing radiographs allow evaluation of coronal plane alignment (e.g., mechanical axis deviation and joint orientation angles), lateral radiographs identify sagittal plane deformities, and advanced imaging such as MRI, CT, or EOS imaging is required for torsional malalignment [18,19]. However, most data derive from XLH, whereas deformities in CR remain poorly characterized, often reported from low-resource settings lacking advanced imaging or standardized terminology.

Despite shared clinical features, CR and HPR differ fundamentally in pathophysiology, remodeling potential, and deformity progression. To address this gap, we systematically reviewed the literature to (i) identify and categorize lower limb deformities across rickets subtypes, (ii) compare their distribution between CR and HPR, and (iii) evaluate the quality and methods of deformity assessment. Our goal is to provide an evidence-based framework to understand rickets-associated deformities and to inform individualized, multidisciplinary care.

## 2. Materials and Methods

A comprehensive literature search was carried out in the PubMed database in order to identify relevant studies published by 4 August 2023, using the Medical Subject Headings (MeSH) terms ‘rickets’ and ‘deformity’ in combination with the Boolean operator ‘AND.’ All records were screened by a single reviewer (A.M.) and the senior authors (A.R. and G.M.) performed random spot checks of the selected publications to ensure consistency and accuracy in the screening process. This review was not prospectively registered in a public database such as PROSPERO (Appendix A).

### 2.1. Study Selection and Eligibility Criteria

Titles and abstracts were screened for relevance. Studies were excluded if they were (1) review articles lacking individual patient data, (2) book chapters, (3) duplicates of patient data, or (4) addressed secondary rickets due to chronic systemic disease (e.g., renal osteodystrophy, inflammatory or dermatologic conditions, or drug-induced rickets). Eligible studies needed to meet the following criteria: (1) English or German language, (2) original data from more than one human patient with rickets and a described lower limb deformity, as well as (3) specification of rickets type or etiology.

### 2.2. Data Extraction, Synthesis and Analysis

A customized data extraction form was developed by the reviewer (A.M.) in consultation with two academic and clinical advisors (A.R. and G.M.) to ensure consistency and data quality. Extracted variables included: first author, study title, total patient number, number of patients with deformities, and number of affected limbs. Potential sources of heterogeneity were explored qualitatively by stratifying results according to rickets subtype (calcipenic vs. hypophosphatemic and further hereditary subgroups). Differences in the distribution of deformity types (varus, valgus, sagittal, torsional, unspecified) were described narratively. Deformities were categorized as varus, valgus, sagittal (e.g., procurvatum/recurvatum), or torsional (e.g., femoral or tibial malrotation). Imprecisely described deformities (e.g., “bowing” or “deformity” without specification) were recorded separately. Diagnostic classification distinguished between calcipenic and hypophosphatemic rickets and further between, calcium deficiency rickets, VDDR, HPR/XLH, ADHR, ARHR and HHRH. Measurement methods (radiological, clinical, gait or undefined) and definitions of deformities (yes/no) were documented. Laboratory parameters (serum calcium, phosphate, alkaline phosphatase) were extracted when available.

### 2.3. Quality and Risk of Bias Assessment

A formal risk of bias assessment was not conducted due to the descriptive and heterogeneous nature of the included studies, which ranged from small observational cohorts to retrospective case series with highly variable methodological quality, reporting standards, and sample sizes. Given the primarily narrative and exploratory aims of this review—namely the mapping and categorization of lower limb deformities in rickets—applying standardized risk of bias tools (e.g., the Newcastle-Ottawa Scale) would not have yielded meaningful or comparable results. Instead, emphasis was placed on the transparent documentation of study characteristics, inclusion criteria, diagnostic approaches, and deformity descriptions in order to allow for qualitative synthesis and interpretation. A formal certainty assessment was not performed. The considerable clinical and methodological heterogeneity among studies further limited the feasibility of a uniform quality scoring.

### 2.4. Statistics

Data analysis was limited to descriptive statistics. Categorical variables were summarized as frequencies and percentages, while continuous variables were reported as means. Inferential statistical methods were not applied. The aim was to provide a descriptive synthesis of the included studies without conducting comparative or predictive analysis. This approach enabled a clear presentation of trends, distributions, and central tendencies within the reported data.

## 3. Results

### 3.1. Study Selection

The process of study identification and selection is outlined in a PRISMA flow diagram in Figure 3. A total of 2056 records were initially retrieved from PubMed. Of these, 227 articles were excluded due to unavailability in either English or German. An additional 387 publications were removed as they were review articles lacking individual patient data, and four records were excluded because full-text access could not be obtained. All remaining potentially eligible full-text articles were retrieved and reassessed for inclusion. Following the removal of duplicates (*n* = 2), studies that were not relevant to topic (*n* = 1201) and single case studies (*n* = 109), a final total of 126 studies met the inclusion criteria and were thus included in this review. All studies included in the analysis are listed in Appendix A and are fully referenced in the bibliography.

### 3.2. Study Characteristics

The 126 studies encompassed a total of 21,568 patients. Across the included studies, 2924 patients with documented lower limb deformities were identified: 1537 with CR and 1387 with HPR. The final analysis comprised 5848 lower extremities, distributed across rickets subtypes. 2630 deformities were documented in calcium deficiency rickets patients, 152 deformities in VDDR patients, 2676 deformities in HPR/XLH patients, 31 in ADHR patients, 34 in ARHR patients, and 22 deformities were reported in patients with HHRH.

### 3.3. Age Reporting

Due to inconsistencies and limited precision in age reporting across the included studies, a pooled mean age could not be calculated. In several studies, age was either reported only for the overall study cohort, without stratification for the subgroup with lower limb deformities, or not provided at the time of deformity assessment. In some cases, only broad age ranges or median ages were mentioned, further limiting comparability. Consequently, a meaningful synthesis of age data was not feasible.

### 3.4. General Findings/Use of Terms

The included studies reported a wide range of lower limb deformities in patients with both calcipenic and hypophosphatemic rickets. Table 1 summarizes the distribution and frequency of these deformities across subtypes and provides an overview of the corresponding deformity numbers.

### 3.5. Unprecise Use of Terms Describing Deformity

A substantial proportion of deformities (32.3%, *n* = 1789) were described non-specifically using terms such as ‘bowing’, ‘deformity’, or ‘twisted legs’ without anatomical precision. This lack of detail limited interpretation and cross-study comparison. Of these, 1137 leg deformities were reported in patients with HPR and 674 in those with CR.

### 3.6. Coronal Plane Deformities

Coronal plane deformities were the most common deformities across all subtypes. Genu varum was described in 2066 limbs (35.3% of all deformities) and genu valgum (*n* = 1199, 21.6% of all deformities) (Figure 4). Genu varum was evenly distributed between CR (*n* = 1075) and HPR (*n* = 991) legs, whereas genu valgum was more prevalent in CR (*n* = 840) than in HPR (*n* = 395) legs. Windswept deformities (consisting of one varus and one valgus leg) were documented in 104 patients, predominantly in the CR group (*n* = 87 patients).

### 3.7. Torsional Deformities

Torsional deformities were predominantly investigated in patients with HPR. Tibial maltorsion was reported in 47 legs (HPR: *n* = 39; CR: *n* = 8), and femoral maltorsion mostly in HPR legs (*n* = 67) and three VDDR legs.

### 3.8. Sagittal Plane Deformities

Sagittal plane deformities have only been assessed in a small number of studies. Sagittal plane deformities of the femur, including procurvatum of the femur (*n* = 85) and recurvatum of the femur (*n* = 6), were exclusively described in HPR/XLH patients. Tibial procurvatum (*n* = 39 in HPR/XLH; *n* = 9 in CR), was described in a higher frequency in the HPR group. No cases of tibial recurvatum were identified.

### 3.9. Quality of Reporting

Diagnostic methods varied widely; only a minority of studies used radiographic parameters such as mechanical axis deviation (MAD), mechanical lateral distal femoral angle (mLDFA), or medial proximal tibial angle (MPTA), while 105 studies (83.3%) relied on clinical inspection alone or did not report their methodology. In total, 79 (62.7%) of the included studies reported on basic laboratory measurements (serum alkaline phosphatase (ALP), phosphate, or calcium) in all included patients. Definition and assessment of functional parameters or inclusion of more than one radiographic parameter (e.g., MAD, mLDFA, MPTA) to quantify deformities has been performed by 22 (17.5%) studies. A total of 20 studies used radiological tools to diagnose deformity, but only two studies used gait analysis to diagnose torsional deformities. Of the included studies, 5.6% (seven out of 126) conducted basic radiological and laboratory assessments while using specific definitions to describe deformities in greater detail, as it can be seen in Figure 5.

The studies that met all three criteria (laboratory tests, deformity assessment, and radiological evaluations) are listed in Table 2. None of the studies including CR patients fulfilled the combination of three suggested quality criteria.

All seven studies that met the above-mentioned quality criteria included exclusively HPR patients with 316 limbs (Figure 6). Among those, genu varum was more frequently reported than genu valgum and femoral maltorsion more often than tibial maltorsion. Sagittal deformities were not reported.

## 4. Discussion

This review is, to our knowledge, the first to synthesize descriptions of lower limb deformities across all rickets subtypes. Our findings highlight significant variability in deformity patterns between HPR and CR, as well as major gaps in reporting quality and methodological robustness. The following analysis addresses key clinical and methodological considerations which have emerged from the systematic examination of 126 studies encompassing 21,568 patients and 5848 lower extremities.

### 4.1. Deformity Patterns in Rickets Subtypes

Sagittal and torsional deformities were predominantly reported in HPR, particularly XLH, whereas windswept and valgus deformities were more frequently described in CR. Whether these observed differences reflect true underlying pathophysiological divergence or result partly from systematic underreporting and methodological limitations in CR remains a critical unresolved question. Persistent torsional deformities in XLH, despite burosumab therapy, support the notion that intrinsic genetic and growth plate abnormalities drive these patterns [10]. This finding aligns with recent prospective studies demonstrating that pharmacological treatment, while correcting biochemical abnormalities, does not uniformly prevent or reverse multiplanar deformities in XLH. Baraka et al. [23] reported that in children with healed nutritional rickets, coronal knee malalignment tends to improve spontaneously, particularly in those younger than eight years and with moderate deformities. In such cases, observation for at least one year may be a reasonable management approach before considering surgical correction. This finding contrasts markedly with HPR, where spontaneous remodeling is less predictable and residual deformities are common. Torsional deformities are rarely quantified in CR, and most studies from low-resource settings—where CR predominates—lack advanced imaging, standardized assessments, or follow-up. This substantial underreporting may obscure true multiplanar deformity prevalence and limit accurate comparisons between rickets subtypes. The lack of longitudinal imaging data makes it impossible to distinguish between primary rickets-related deformities, residual skeletal changes from previous episodes of CR or from other comorbidities, and specific dysplasia-like features in genetic conditions such as XLH. Longitudinal studies incorporating standardized biochemical, radiological, and orthopedic assessments are needed to clarify these mechanisms and track deformity evolution.

### 4.2. Reporting Quality and Methodological Challenges

Deformity reporting across studies was inconsistent and often lacked precision, as many authors used nonspecific terms such as “bowing” instead of providing quantitative measurements. The subgroup that included the largest number of imprecise descriptions is HPR/XLH, which implies that deformity characterization in this patient cohort remains inconsistent and insufficiently standardized. This is particularly problematic given that XLH is the most common form of HPR but involves specific skeletal features. Only 5.6% of all 126 included studies systematically documented deformities with standardized radiological and laboratory parameters. Several studies demonstrated a clear disciplinary bias in their reporting focus. This disciplinary divide represents a major barrier to comprehensive understanding of rickets-related deformities. Pediatric endocrinology-focused publications typically provided detailed biochemical and genetic characterization but superficial musculoskeletal descriptions, while orthopedic studies offered precise radiographic analyses but lacked laboratory data, diagnoses, or genetic confirmation. This disciplinary divide underscores the need for multidisciplinary collaboration in order to achieve comprehensive phenotyping that integrates metabolic, radiological, and clinical data.

Historically, XLH was often described under umbrella terms such as ‘vitamin D-resistant rickets’ or ‘hypophosphatemic rickets’ without further etiological specification until the genetic basis and phosphate-wasting pathophysiology were clarified [24]. As a result, many studies failed to distinguish XLH from other causes of HPR, leading to heterogeneous cohorts. This nosological confusion, which persisted for decades before genetic testing became available, has created a legacy of diagnostic uncertainty in the literature. Furthermore, genetic testing was not widely available in earlier reports, meaning that many diagnoses were based solely on biochemical profiles or clinical presentation. These inconsistencies hinder precise subtype differentiation and limit the ability to perform accurate subgroup analysis in this review. The interpretation of observed deformity patterns should be considered in light of potential confounding factors. Variations in patient age of onset and deformity description, disease stage, and treatment exposure, as well as differences in diagnostic confirmation and imaging methodology, may have influenced the reported distributions. Although these aspects were addressed indirectly throughout the review, their inconsistent documentation across studies limits the ability to determine the extent of their impact.

An additional challenge is distinguishing physiological alignment changes in children from pathological deformities. Normal development includes transient varus and valgus phases; persistence or progression beyond age-appropriate norms, especially in the context of biochemical abnormalities (hypocalcemia, hypophosphatemia, elevated ALP), should prompt clinical evaluation for rickets [25]. The majority of included CR studies did not provide age-matched normative data or reference values for radiographic parameters, making it difficult to assess whether reported deformities exceed expected developmental variation.

The lack of longitudinal studies with standardized radiographic and clinical follow-up limits our understanding of how deformities evolve and respond to medical or surgical interventions. This interpretation is further complicated by the fact that few studies clearly differentiate primary deformities from iatrogenic changes resulting from previous surgical procedures. When surgical history is reported, key contextual details—such as the patient’s age at the time of surgery, the specific limb segment involved, or the nature of the intervention—are reported only infrequently and without adequate detail. Without this information, it is difficult to distinguish between the natural history of a deformity and treatment-related sequelae. As these variables were not systematically extracted in the present study, a quantitative assessment was not possible. Standardized reporting of surgical history, including patient age, anatomical site, procedure type, and follow-up duration, would substantially improve interpretability and facilitate meaningful comparisons across studies.

Given these methodological inconsistencies and the predominance of descriptive data, the present review relied on descriptive statistical summaries rather than inferential analyses. While this approach provides insights into the distribution and frequency of reported deformities, it does not allow for statistical testing of significance or robustness. Future systematic reviews based on standardized and quantitatively comparable datasets should incorporate inferential and sensitivity analyses to strengthen analytical depth and interpretability.

### 4.3. Pathophysiological and Clinical Implications

The observed differences in deformity patterns align with the concept that the persistence of deformities in XLH arises from intrinsic growth-plate defects and impaired remodeling, whereas deformities in CR tend to resolve following correction of the underlying metabolic deficiency. This has direct implications for management: while deformities in CR have been shown to remodel completely with timely pharmacological correction of the underlying deficiency, HPR—despite pharmacological treatment such as conventional supplementation or burosumab—has been shown to result in residual deformities, making surgical correction necessary in a subset of patients [10,23]. However, it remains uncertain whether these differences truly reflect underlying pathophysiology or are partly due to underreporting in CR, underscoring the need for further investigation. Future studies should record deformities with precise timing relative to diagnosis and document their response to therapy, using standardized radiographic monitoring and functional assessments (e.g., gait analysis) in order to inform the optimal timing and type of intervention.

### 4.4. Limitations

This review has several limitations. Firstly, it is substantially limited by heterogeneity in study designs (ranging from single-case reports to observational cohorts), inconsistent and imprecise deformity reporting, variable age ranges, and widely differing treatment exposures and follow-up durations, all of which make meaningful subgroup comparisons and meta-analysis infeasible. The vast heterogeneity in measurement techniques, outcome definitions, and baseline characteristics across studies prevents pooling of quantitative data and necessitates reliance on qualitative synthesis. Furthermore, the lack of a formal risk of bias assessment represents an additional limitation of this review. However, as detailed in the Section 2, the application of such tools to predominantly small case reports and retrospective cohorts without standardized diagnostic confirmation would have yielded uniformly low or indeterminate quality scores with limited interpretative value. Instead, transparent documentation of study characteristics provides readers with explicit information to assess the credibility of individual studies. The literature search was conducted exclusively in PubMed, which was selected for its comprehensive biomedical coverage and standardized indexing using MeSH terms. Nevertheless, this approach may have excluded studies indexed in other databases, and the potential for publication bias cannot be completely ruled out. Language restrictions (English/German) may have excluded relevant data, particularly from low-resource settings. All records were screened by a single reviewer, which may introduce selection bias despite predefined inclusion criteria and senior author oversight. A further limitation of this study is its retrospective design and the absence of protocol registration. The predominance of small retrospective case series and single-center reports limits the certainty of evidence and the ability to draw firm conclusions regarding etiologic and pathophysiologic differences between CR and HPR.

### 4.5. Future Research Directions

Future research should prioritize prospective, standardized documentation of lower limb deformities across all rickets subtypes, using protocol-driven standards that integrate multidisciplinary expertise. This should include multiplanar imaging and functional assessment, complemented by basic radiological parameters such as MAD, mLDFA, MPTA, and torsion angles. Clear definitions of deformity types must be explicitly stated to ensure consistency, and genetic confirmation for hereditary forms are critical. Ultimately, such data are essential for defining disease-specific deformity patterns, optimizing timing of orthopedic intervention, and improving long-term outcomes. Phenotype-genotype correlation studies based on reproducible assessment standards will be essential for understanding how specific genetic variants influence deformity manifestations and skeletal remodeling.

We propose a minimal quality framework to guide future studies and advanced research and to support evidence-based, subtype-specific management. This framework emphasizes standardized radiological measurement using validated parameters, comprehensive biochemical profiling, and explicit deformity-specific terminology to enable reproducibility, comparability, and synthesis across future studies. Adherence to this framework will substantially improve the evidence base and enable future high-quality systematic reviews incorporating meta-analysis where appropriate (Figure 7).

Gait analysis has been increasingly used to assess gait deviations and lower limb deformities in rare bone diseases [26]. It can improve our understanding of leg deformities in rickets; however, it is an expensive examination and limited to a few specialized centers. New marker-free gait analysis tools could offer a feasible alternative in low-income settings to improve our research of rickets and lower limb deformities.

## 5. Conclusions

This review highlights the high prevalence of and phenotypic variability in lower limb deformities in pediatric rickets. While genu varum and genu valgum were commonly reported across both calcipenic and hypophosphatemic forms, axial and sagittal deformities, particularly torsional malalignment and procurvatum, were predominantly observed in HPR, mainly in XLH patients. In contrast, such deformities seem under-reported or are poorly described in calcipenic forms, suggesting a potential gap in clinical recognition or diagnostic assessment.

The findings reveal significant inconsistencies in diagnostic terminology, imaging standards, and deformity classification across studies, which limits data comparability and impedes the development of unified treatment strategies. Only a small proportion of studies employed rigorous radiological definitions and comprehensive diagnostic protocols.

To improve clinical care and research in this field, future studies should adopt standardized definitions and objective measurement tools for skeletal deformities, including in transversal and sagittal planes. Moreover, longitudinal data are needed to evaluate deformity progression and the skeletal response to both medical and surgical interventions. Understanding the disease-specific patterns of skeletal involvement is essential for optimizing outcomes and individualizing care for patients affected by rickets.

## Figures and Tables

**Figure 1 jcm-14-08586-f001:**
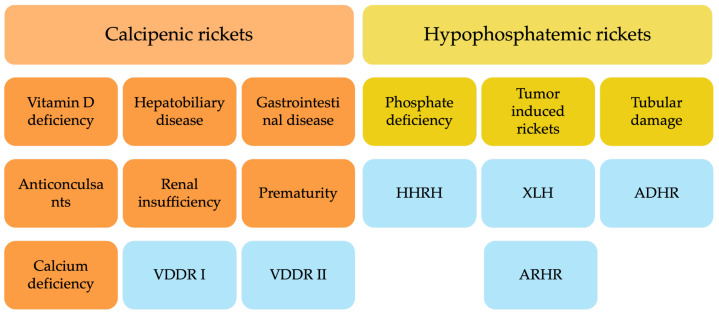
Classification of calcipenic rickets (CR) and hypophosphatemic rickets (HPR); hereditary forms of rickets are illustrated in blue. Vitamin D-dependent rickets (VDDR) autosomal dominant hypophosphatemic rickets (ADHR), hereditary hypophosphatemic rickets with hypercalciuria (HHRH), autosomal recessive hypophosphatemic rickets (ARHR), and X-linked hypophosphatemia (XLH).

**Figure 2 jcm-14-08586-f002:**
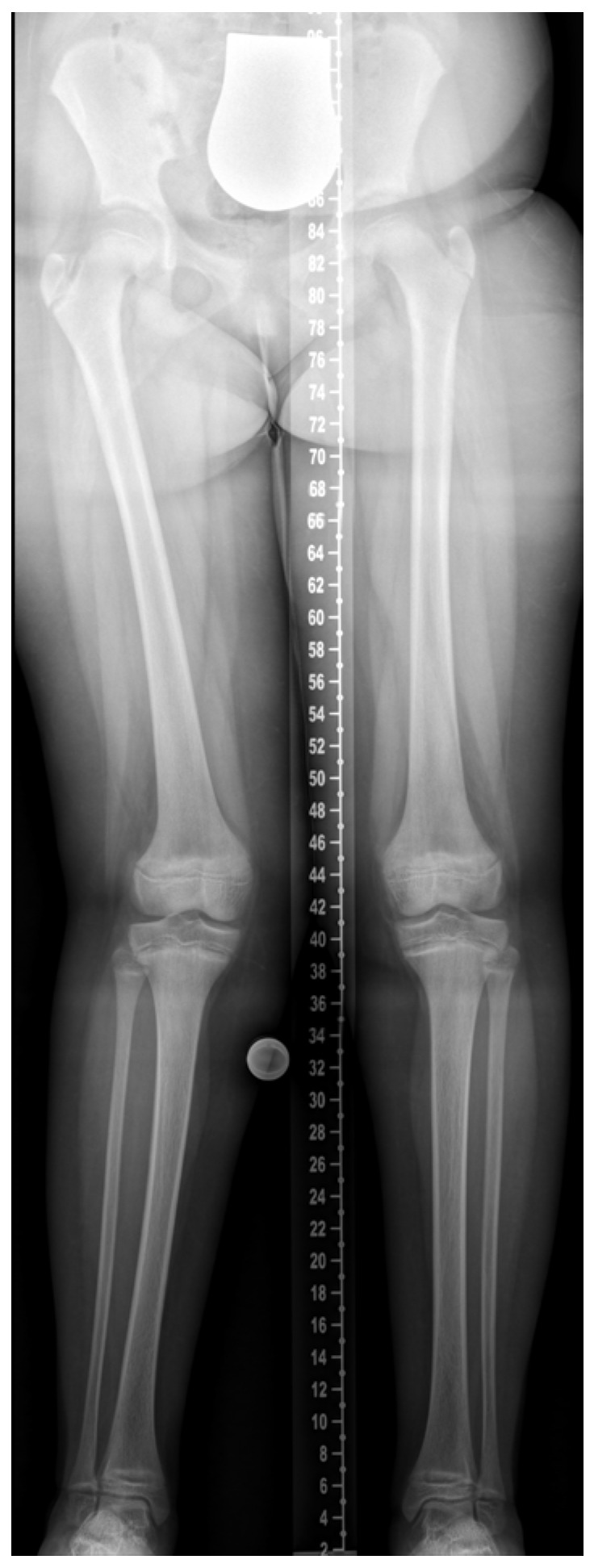
A 10-year-and-10-month-old girl with nutritional rickets. Radiographic findings show valgus deformity of the right leg and varus deformity of the left leg (windswept deformity). Image courtesy of Dr. Gabriel T. Mindler, Department of Pediatric Orthopedics, Orthopedic Hospital Speising, Vienna, Austria.

**Figure 3 jcm-14-08586-f003:**
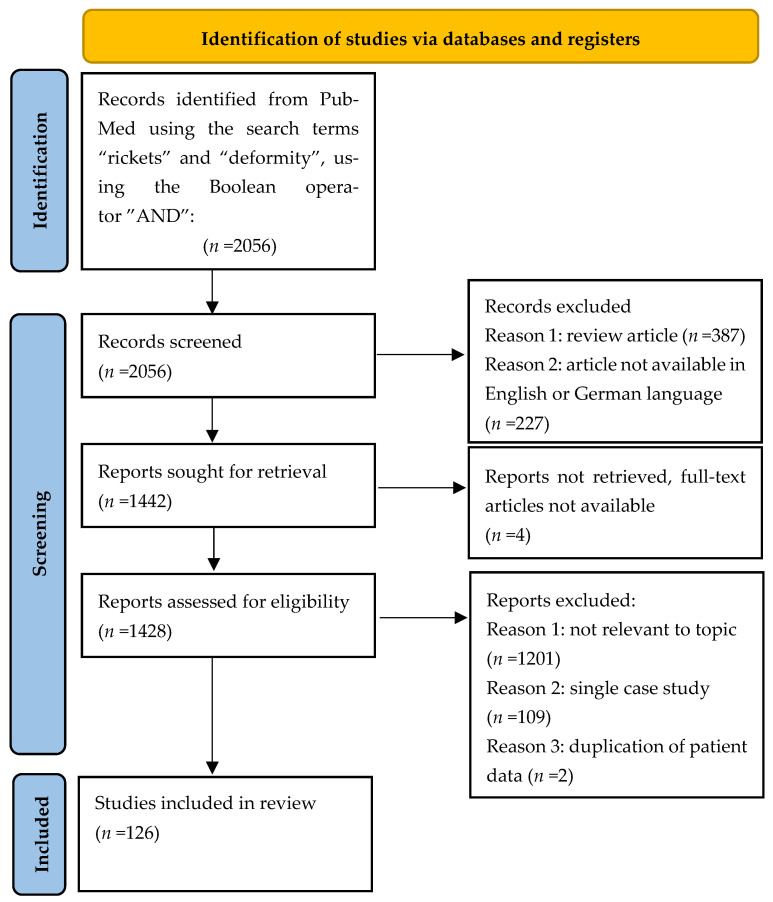
PRISMA (Preferred Reporting Items for Systematic Reviews and Meta-Analyses) 2020 diagram for study identification and selection.

**Figure 4 jcm-14-08586-f004:**
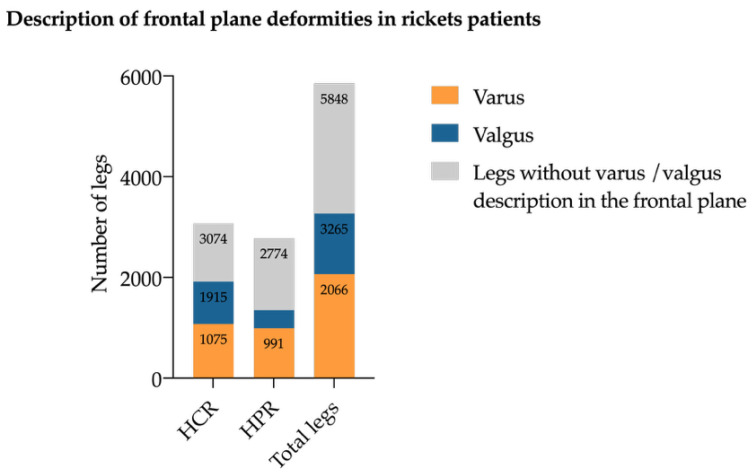
Number of frontal plane deformities. Varus (illustrated in orange) and valgus (illustrated in blue) deformity in legs of CR (calcipenic) and HPR (hypophosphatemic rickets) patients. Legs without varus or valgus deformities are illustrated in gray.

**Figure 5 jcm-14-08586-f005:**
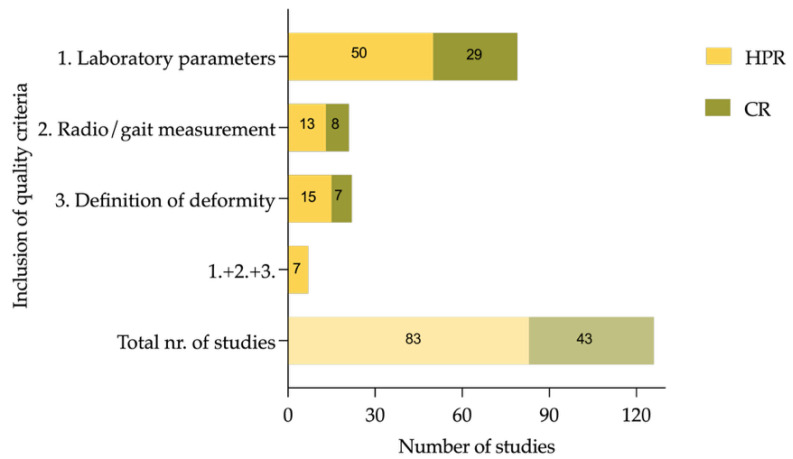
Number of studies that fulfill minimal quality criteria: (1) studies including laboratory parameters (alkaline phosphatase (ALP) and phosphate or calcium) of all the patients described, (2) studies using radiological measurement of deformity (e.g., mechanical axis deviation (MAD), (3) studies defining deformities, (4) studies that fulfill all three before mentioned quality criteria, and (5) total number of studies assessed in this analysis.

**Figure 6 jcm-14-08586-f006:**
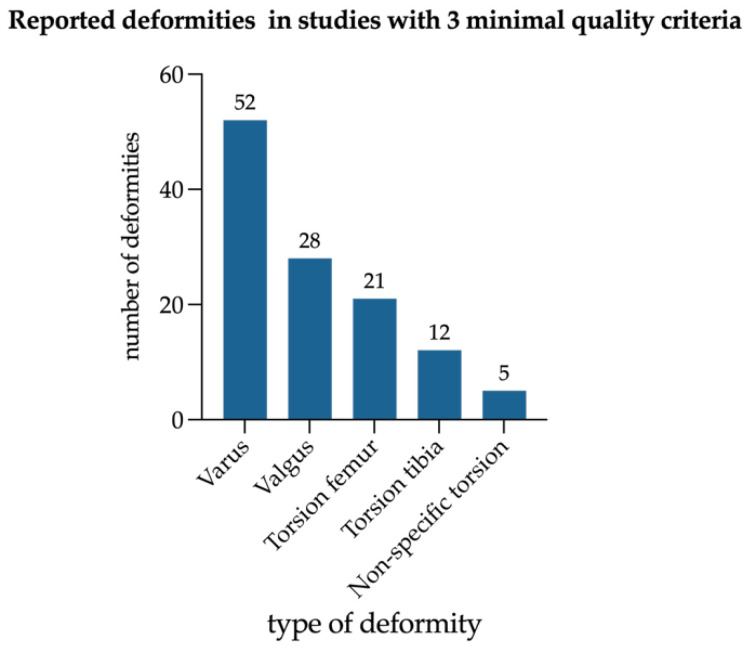
Percentage of reported deformities in total legs of seven studies fulfilling three quality criteria, no sagittal deformities in femur or tibia were reported in such. No study reporting CR (calcipenic rickets) patients fulfilled the suggested quality criteria.

**Figure 7 jcm-14-08586-f007:**
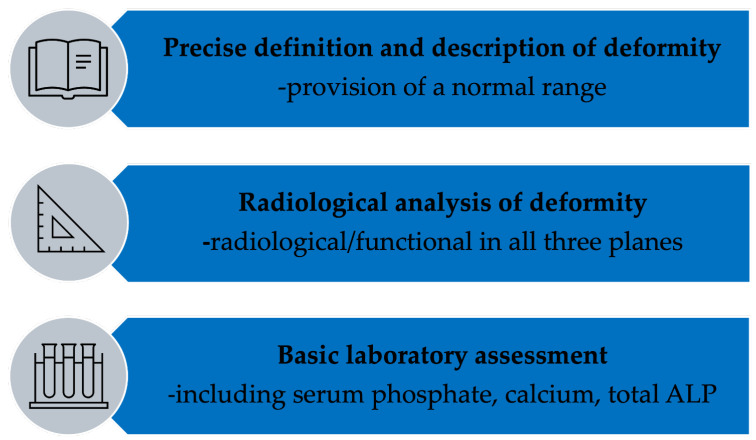
Suggested minimal quality criteria for studies on lower limb deformities in the context of bone disorders. ALP = alkaline phosphatase.

**Table 1 jcm-14-08586-t001:** Key findings of reported leg deformity per leg in 2924 patients: of those 1537 patients with calcipenic rickets (CR), 1387 patients had hypophosphatemic rickets (HPR).

	CR		HPR			
	Calcium Deficiency Rickets	VDDR	HPR/XLH	ADHR	ARHR	HHRH
Count of deformities	2630	153	2670	31	34	22
Varus	1010	65	972	1	16	2
Valgus	777	63	340	1	8	10
Windswept	86 *	1 *	16 *	1 *	0	0
Maltorsion femur	0	3	67	0	0	0
Maltorsion tibia	6	2	39	0	0	0
Femoral procurvatum	0	0	85	0	0	0
Femoral recurvatum	0	0	6	0	0	0
Tibial procurvatum	7	2	39	0	0	0
Unprecise term for deformity	658	16	1090	27	10	10

VDDR = Vitamin D-dependent rickets, XLH = X-linked hypophosphatemia, ADHR = autosomal dominant hypophosphatemic rickets, ARHR = autosomal recessive hypophosphatemic rickets, HHRH = hereditary hypophosphatemic rickets with hypercalciuria, * = number of patients.

**Table 2 jcm-14-08586-t002:** Studies meeting all three criteria: laboratory testing (serum alkaline phosphatase (ALP), phosphate or calcium), deformity assessment, and radiological evaluations.

Choi et al. (2002) [20]
Petje et al. (2008) [21]
Song et al. (2014) [22]
Mindler et al. (2020) [14]
Mindler et al. (2022) [10]
Bonnet-Lebrun et al. (2023) [12]
Bonnet-Lebrun et al. (2023) [11]

## Data Availability

All the described data can be seen in Appendix A.

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
