# Peer review of "Lower Limb Deformity in Different Types of Rickets"

_jcm, 2025, doi:10.3390/jcm14238586_

Round 1
Reviewer 1 Report
Comments and Suggestions for Authors
Dear Authors,
Your manuscript entitled “Lower limb deformity in different types of rickets – A systematic
literature review” addresses a relevant and valuable topic. The review successfully
summarizes deformity patterns in different etiologies of rickets and underlines the need for
improved reporting standards. The methodological approach is sound, but several minor
revisions are required regarding structure, figure referencing, consistency, and language.
Below, I provide detailed feedback structured by section.
General Comments
• The PRISMA-based structure and clinical relevance are strengths of this paper.
• However, the manuscript contains numerous formatting, orthographic, and
typographical inconsistencies (spacing, capitalization, abbreviation redundancy,
decimal separators, etc.).
• Please use American English consistently (e.g., pediatric, etiology, customized).
• Ensure that all figures and tables are sequentially numbered and referenced in the text.
• Avoid repeating abbreviations (CR, HPR, VDDR, XLH, MAD, MPTA, mLDFA) after
first definition.
• Verify the visual quality and caption consistency of all figures (some are low
resolution, others italicized or in bold).
• Delete all red underlines, markup traces, or inconsistent font styles.
Methodological and Content-Related Issues
1. Clarify who performed the screening and data extraction (“single reviewer”) and
whether any verification was done.
2. The literature search used only PubMed – please justify this limitation explicitly.
3. Although you mention that a formal bias assessment was not possible, a brief
qualitative summary (e.g., by study type or reporting completeness) should be added.
4. Ensure each figure is cited in the main text (Figures 1, 4, 6 currently are not).
5. Define all abbreviations in each figure/table legend (CR, HPR, ALP, MAD, etc.).
6. Simplify the discussion by removing repetitive statements about methodological
limitations.
7. In Figure 7 (“Minimal Quality Framework”), unify font size and remove unnecessary
bold bullet points.
Detailed Formal and Orthographic Corrections
Abstract
• “paediatric disorder” → “pediatric disorder”
• Define abbreviation PRISMA (Preferred Reporting Items for Systematic Reviews and
Meta-Analyses).
• Be consistent in using commas for thousands (e.g., 21,000 patients).
Introduction
• Add missing spaces before reference brackets (e.g., “rickets[3]” → “rickets [3]”).
• “CR most frequently…” instead of “Calcipenic rickets most frequently…”
• Insert missing spaces in “1A (VDDR1A)” and “2A (VDDR2A)”.
• Use abbreviations consistently (“HPR” instead of “Hypophosphatemic rickets”).
• “conditions [6][7]” instead of “conditions[6][7]”.
• “vitamin D synthesis [8]” instead of “vitamin D synthesis[8]”.
• “phosphate alone[9]” → “phosphate alone [9]”.
• Figure 1 is not referenced in the text—please cite.
• Caption Figure 1: spell out abbreviations “CR” and “HPR”.
• “... [10][11][12][13][14]” instead of “… [10][11][12][13] [14]”.
• Do not format “Figure 2” in bold; do not italicize captions.
• Remove redundant caption text (“Figure 2 caption repeated in main text”).
• Add missing spaces before references throughout.
Materials and Methods
• Identify who the “single reviewer” was and clarify verification.
• “customised” → “customized”.
• Use lowercase for headings: “risk of bias assessment” instead of “Risk of Bias
Assessment”.
• “CR and HPR” instead of “calcipenic and hypophosphatemic rickets” (use
abbreviations).
• Delete red underlines and markup.
Results
• “Figure 1” → “Figure 3” (correct numbering).
• Maintain consistent number formatting (no spaces before/after “=”). Examples:
o “(n = 2)” → “(n=2)”,
o “(n = 1201)” → “(n=1201)”.
• Use periods for decimals (83.3 %, 62.7 %, 17.5 %).
• Define ALP earlier in text.
• “phophorus” → “phosphate”.
• “≥ 1” → “≥1” (delete extra space).
• Use “MAD” instead of “mechanical axis deviation” (already defined).
• “MPTA” instead of “mMPTA”.
• In Figure 3, ensure no spaces around “=” in all values.
• Abbreviations CR, HPR, VDDR, HHRH already defined—avoid redefinition.
• Add short introductory sentences before tables that currently follow subheadings
without text.
• Table 1: define VDDR, HPR/XLH, ADHR, ARHR, HHRH in legend; do not redefine
CR or HPR.
• Figure 4: must be cited in text; adjust color legend (Varus = yellow, Valgus = blue,
Neutral = grey).
• Figure 4 caption: define abbreviations CR and HPR; “HPR: n=39; CR: n=8” (no
spaces around “=”).
• Figure 5 caption: define ALP, CR, HPR, MAD; “phosphate” instead of “Phophorus”;
“calcium” instead of “Calcium”.
• Table 2 legend: define ALP and HPR; use “phosphate”. Remove study titles/journal
names (redundant).
• Figure 6: not cited in text; labeling poor quality; use “Varus”/“Valgus” instead of
“Vara”/“Valga”; improve resolution.
• Figure 6 caption: define “CR”.
Discussion
• Use lowercase in section titles (“Deformity patterns in rickets subtypes”, “Reporting
quality and methodological challenges”).
• “patterns [10]” instead of “patterns[10]”; “correction [24]” instead of “correction[24]”.
• “patients [10][24].” instead of “patients.10][24]”.
• Delete extra space in “studies/ case reports” → “studies/case reports”.
• “has been shown residual deformities” → “has been shown to result in residual
deformities.”
• Clarify “o” → “or/to”.
• “phosphorus” → “phosphate” throughout.
• Figure 7: define ALP; bullet points in middle and lower blue blocks should not be
bold.
• Add missing closing parenthesis.
• Ensure correct figure numbering (now “Figure 7”).
Conclusions
• Do not redefine abbreviations already listed.
Abbreviations Section
• Unify font sizes and fix inappropriate line breaks.
• Ensure consistent formatting for “VDDR1A”, “VDDR2A”, “mLDFA”, “PRISMA”.
Appendix A
• Section title: “Study Quality criteria” → “Study quality criteria” (sentence case).
Additional Recommendations
1. Conduct a final language and formatting edit by a native English editor.
2. State explicitly that only PubMed was searched and only one reviewer screened
studies—include in “Limitations”.
3. Move key results tables from Appendix to main text for better readability.
4. Ensure all figures are ≥ 300 dpi and captions meet JCM format requirements.
Overall Recommendation: Minor Revision

The English language is generally understandable but requires moderate editing to meet JCM
standards. Specific issues include:
• Inconsistent use of British vs. American spelling (paediatric/pediatric,
customised/customized, aetiology/etiology).
• Numerous missing spaces before citation brackets and inconsistent punctuation.
• Redundant abbreviation definitions; abbreviations sometimes inconsistently
capitalized.
• Decimal separators should use periods (83.3 %) instead of commas (83,3 %).
• Occasional grammatical errors (“has been shown residual deformities” → “has been
shown to result in residual deformities”).
• Non-standard capitalization in section titles and figure captions.
A final language and style review by a professional editor is strongly recommended
before acceptance.
Author Response
RESPONSE TO REVIEWER 1
Dear Authors,
Your manuscript entitled “Lower limb deformity in different types of rickets – A systematic
literature review” addresses a relevant and valuable topic. The review successfully
summarizes deformity patterns in different etiologies of rickets and underlines the need for
improved reporting standards. The methodological approach is sound, but several minor
revisions are required regarding structure, figure referencing, consistency, and language.
Below, I provide detailed feedback structured by section.
General Comments
• The PRISMA-based structure and clinical relevance are strengths of this paper.
Response to Reviewer 1: Thank you, Reviewer 1, for the thorough revision of our manuscript. We have incorporated this detailed list point by point into our manuscript and hope this fully meets your expectations. Once again, thank you for the time and effort you invested in helping us improve our manuscript.
1.1 However, the manuscript contains numerous formatting, orthographic, and
typographical inconsistencies (spacing, x, abbreviation redundancy,
decimal separators, etc.).
• Please use American English consistently (e.g., pediatric, etiology, customized).
• Ensure that all figures and tables are sequentially numbered and referenced in the text.
• Avoid repeating abbreviations (CR, HPR, VDDR, XLH, MAD, MPTA, mLDFA) after
first definition.
• Verify the visual quality and caption consistency of all figures (some are low
resolution, others italicized or in bold).
• Delete all red underlines, markup traces, or inconsistent font styles.
Methodological and Content-Related Issues
Response 1.1: Thank you for the thorough revision. We have implemented the changes accordingly. We have also had the entire manuscript reviewed by an English medical proofreader to meet the reviewer’s expectations.
1.2. Clarify who performed the screening and data extraction (“single reviewer”) and
whether any verification was done.
Response 1.2: We thank the reviewer for this helpful comment. In the revised version, we explicitly state that all records were screened by a single reviewer (AM). This information has been added to the "Methods" section. Furthermore, we have added that the senior authors (AR, GM) performed spot checks of the selected publications to ensure consistency and accuracy in the screening process.
1.3. The literature search used only PubMed – please justify this limitation explicitly.
Response 1.3: We acknowledge that limiting the search to PubMed may restrict the comprehensiveness of the literature retrieval. PubMed was selected as it provides the most systematically indexed and widely used biomedical database, covering nearly all peer-reviewed journals relevant to pediatric bone and mineral disorders. Given the focused clinical scope of this review and the standardized indexing of medical subject headings (MeSH) in PubMed, we considered this approach appropriate and methodologically consistent. This rationale and limitation have now been explicitly stated in the revised manuscript. In addition, we have changed the term "systematic literature review" to "literature review" in both the title and the manuscript to better meet the expectations of all three reviewers and more accurately reflect our methodology.
1.4 Although you mention that a formal bias assessment was not possible, a brief
qualitative summary (e.g., by study type or reporting completeness) should be added.
Response 1.4: We appreciate this constructive suggestion. We have now added a qualitative summary in the Methods and Results sections describing the heterogeneity of study designs (ranging from small case reports to retrospective case series and observational cohorts) and the variability in reporting completeness. Specifically, we note that only 5.6% of studies met our predefined quality criteria (laboratory parameters, radiological measurements, and deformity definitions), with significant disciplinary bias evident—pediatric endocrinology publications typically provided detailed biochemical characterization but superficial musculoskeletal descriptions, while orthopedic studies offered precise radiographic analyses but lacked comprehensive laboratory data. This qualitative assessment has been integrated to provide readers with context regarding the quality and limitations of the evidence base.
1.5 . Ensure each figure is cited in the main text (Figures 1, 4, 6 currently are not).
Response 1.5. Thank you for identifying this oversight. All figures are now cited sequentially in the main text.
1.6. Define all abbreviations in each figure/table legend (CR, HPR, ALP, MAD, etc.).
Response 1.6. All abbreviations are now defined in each figure and table legend to ensure that each visual element is self-contained and interpretable without reference to the main text.
1.7 . Simplify the discussion by removing repetitive statements about methodological
Limitations.
Response 1.6.Thank you for this comment. We have carefully revised the Discussion section to improve readability and focus. Repetitive statements concerning methodological limitations have been removed or merged for conciseness. The revised text now presents the limitations more succinctly and avoids redundancy while maintaining essential methodological context.
1.8 . In Figure 7 (“Minimal Quality Framework”), unify font size and remove unnecessary
bold bullet points.
Response 1.8. Figure 7 has been revised with consistent font sizes and formatting. Unnecessary bold formatting in bullet points has been removed for improved visual clarity.
1.9. Detailed Formal and Orthographic Corrections
Response 1.9: Thank you. We made changes accordingly.
Abstract
• “paediatric disorder” → “pediatric disorder”
• Define abbreviation PRISMA (Preferred Reporting Items for Systematic Reviews and
Meta-Analyses).
• Be consistent in using commas for thousands (e.g., 21,000 patients).
Introduction
• Add missing spaces before reference brackets (e.g., “rickets[3]” → “rickets [3]”).
• “CR most frequently…” instead of “Calcipenic rickets most frequently…”
• Insert missing spaces in “1A (VDDR1A)” and “2A (VDDR2A)”.
• Use abbreviations consistently (“HPR” instead of “Hypophosphatemic rickets”).
• “conditions [6][7]” instead of “conditions[6][7]”.
• “vitamin D synthesis [8]” instead of “vitamin D synthesis[8]”.
• “phosphate alone[9]” → “phosphate alone [9]”.
• Figure 1 is not referenced in the text—please cite.
• Caption Figure 1: spell out abbreviations “CR” and “HPR”.
• “... [10][11][12][13][14]” instead of “… [10][11][12][13] [14]”.
• Do not format “Figure 2” in bold; do not italicize captions.
• Remove redundant caption text (“Figure 2 caption repeated in main text”).
• Add missing spaces before references throughout.
Materials and Methods
• Identify who the “single reviewer” was and clarify verification.
• “customised” → “customized”.
• Use lowercase for headings: “risk of bias assessment” instead of “Risk of Bias
Assessment”.
• “CR and HPR” instead of “calcipenic and hypophosphatemic rickets” (use
abbreviations).
• Delete red underlines and markup.
Results
• “Figure 1” → “Figure 3” (correct numbering).
• Maintain consistent number formatting (no spaces before/after “=”). Examples:
o “(n = 2)” → “(n=2)”,
o “(n = 1201)” → “(n=1201)”.
• Use periods for decimals (83.3 %, 62.7 %, 17.5 %).
• Define ALP earlier in text.
• “phophorus” → “phosphate”.
• “≥ 1” → “≥1” (delete extra space).
• Use “MAD” instead of “mechanical axis deviation” (already defined).
• “MPTA” instead of “mMPTA”.
• In Figure 3, ensure no spaces around “=” in all values.
• Abbreviations CR, HPR, VDDR, HHRH already defined—avoid redefinition.
• Add short introductory sentences before tables that currently follow subheadings
without text.
• Table 1: define VDDR, HPR/XLH, ADHR, ARHR, HHRH in legend; do not redefine
CR or HPR.
• Figure 4: must be cited in text; adjust color legend (Varus = yellow, Valgus = blue,
Neutral = grey).
• Figure 4 caption: define abbreviations CR and HPR; “HPR: n=39; CR: n=8” (no
spaces around “=”).
• Figure 5 caption: define ALP, CR, HPR, MAD; “phosphate” instead of “Phophorus”;
“calcium” instead of “Calcium”.
• Table 2 legend: define ALP and HPR; use “phosphate”. Remove study titles/journal
names (redundant).
• Figure 6: not cited in text; labeling poor quality; use “Varus”/“Valgus” instead of
“Vara”/“Valga”; improve resolution.
• Figure 6 caption: define “CR”.
Discussion
• Use lowercase in section titles (“Deformity patterns in rickets subtypes”, “Reporting
quality and methodological challenges”).
• “patterns [10]” instead of “patterns[10]”; “correction [24]” instead of “correction[24]”.
• “patients [10][24].” instead of “patients.10][24]”.
• Delete extra space in “studies/ case reports” → “studies/case reports”.
• “has been shown residual deformities” → “has been shown to result in residual
deformities.”
• Clarify “o” → “or/to”.
• “phosphorus” → “phosphate” throughout.
• Figure 7: define ALP; bullet points in middle and lower blue blocks should not be
bold.
• Add missing closing parenthesis.
• Ensure correct figure numbering (now “Figure 7”).
Conclusions
• Do not redefine abbreviations already listed.
Abbreviations Section
• Unify font sizes and fix inappropriate line breaks.
• Ensure consistent formatting for “VDDR1A”, “VDDR2A”, “mLDFA”, “PRISMA”.
Appendix A
• Section title: “Study Quality criteria” → “Study quality criteria” (sentence case).
Additional Recommendations
1. Conduct a final language and formatting edit by a native English editor.
2. State explicitly that only PubMed was searched and only one reviewer screened
studies—include in “Limitations”.
3. Move key results tables from Appendix to main text for better readability.
4. Ensure all figures are ≥ 300 dpi and captions meet JCM format requirements.
Overall Recommendation: Minor Revision
Response 1.9: Thank you. We have systematically addressed all detailed corrections listed. Regarding the Reviewer’s suggestion to move the results tables to the main text, we appreciate the importance of ensuring that key results are accessible to the reader. However, we have retained these tables in the appendix because their inclusion in the main text would substantially exceed the permitted manuscript length. To enhance readability, we have clearly referenced the location of these tables in the Results section so that interested readers can locate them easily in the appendix. We respectfully believe this approach maintains clarity within the body text while conforming to journal length requirements. If the other Reviewers and/or the Editor would consider it as beneficial to move the tables, we can adapt the manuscript accordingly.
1.10 Comments on the Quality of English Language
The English language is generally understandable but requires moderate editing to meet JCM
standards. Specific issues include:
• Inconsistent use of British vs. American spelling (paediatric/pediatric,
customised/customized, aetiology/etiology).
• Numerous missing spaces before citation brackets and inconsistent punctuation.
• Redundant abbreviation definitions; abbreviations sometimes inconsistently
capitalized.
• Decimal separators should use periods (83.3 %) instead of commas (83,3 %).
• Occasional grammatical errors (“has been shown residual deformities” → “has been
shown to result in residual deformities”).
• Non-standard capitalization in section titles and figure captions.
A final language and style review by a professional editor is strongly recommended
before acceptance.
Response 1.10.Thank you for this helpful comment. The entire manuscript was thoroughly reviewed and edited for linguistic accuracy, consistency in American English spelling, and overall readability. Abbreviations, decimal formatting, and capitalization have been standardized throughout.In addition, the entire manuscript underwent a final language review by an English medical proofreader to ensure clarity and conformity with JCM standards.All changes are highlighted in the marked-up version of the new manuscript.
Reviewer 2 Report
Comments and Suggestions for Authors
I read with interest the manuscript titled "Lower limb deformity in different types of rickets".
The introduction is divided into subsections with figures and tables, which is not recommended. I ask that the introduction be a comprehensive background of the topic under investigation, without subsections, with a clear statement of the aim of the systematic review.
It is not common for a systematic review to be based on only one database. For publication in a high-quality journal such as this, it is of utmost importance that the systematic review be based on multiple databases and registered.
For systematic reviews, it is not recommended that all records be reviewed by a single reviewer. Please check how scientifically based systematic reviews are written and conducted.
Systematic tools such as the Newcastle-Ottawa Scale or others are designed to accommodate a broad range of observational study designs, including case series and cohorts, and can often be adapted or supplemented to account for heterogeneity. These assessments help elucidate the methodological strengths and weaknesses of individual studies, guiding readers in interpreting the robustness of the evidence base. Even in reviews with primarily narrative or exploratory aims, including a transparent risk of bias assessment enhances the transparency of findings and supports nuanced interpretation. It also facilitates identification of potential biases that could influence reported outcomes or deformity descriptions. While heterogeneity poses challenges, it does not preclude the application of systematic quality appraisal methods; rather, it underscores the importance of contextualizing bias assessments within the variability of study designs and quality.
While descriptive statistics provide valuable insights into the data's distribution and trends, relying exclusively on this approach may limit the depth of your analysis. Without inferential statistical methods, it is difficult to assess the significance of observed differences or relationships, which could lead to overinterpretation of apparent patterns. Additionally, the absence of sensitivity analyses means potential robustness or variability of the findings remains unexamined. Incorporating inferential techniques and sensitivity analyses could enhance the rigor and interpretability of your results.
I would like to suggest that the results section currently appears somewhat simplistic, as it primarily reports findings without discussing the potential influence of confounding factors. In line with established reporting guidelines such as PRISMA, it is important to consider and address possible confounders that could impact the validity and interpretation of your findings. Including an analysis or discussion of these factors would greatly enhance the scientific rigor of your review and provide a more nuanced understanding of the results.
While there may be additional limitations not recognized, the current identified factors—such as heterogeneity in study designs, inconsistent reporting, lack of bias assessment, language restrictions, and the predominance of retrospective studies—are substantial enough to significantly compromise the ability to draw scientifically sound conclusions from this systematic review.
Author Response
RESPONSE TO REVIEWER 2
Comments and Suggestions for Authors
I read with interest the manuscript titled "Lower limb deformity in different types of rickets".
We thank the reviewer for this comprehensive and constructive feedback. We acknowledge the limitations inherent to our analytical approach and methodology and have implemented several revisions accordingly.
The introduction is divided into subsections with figures and tables, which is not recommended. I ask that the introduction be a comprehensive background of the topic under investigation, without subsections, with a clear statement of the aim of the systematic review.
Response 1. We have revised the structure of the Introduction to provide a unified, continuous background without subsections and with a clear presentation of the aims, as recommended. Figures and tables have been repositioned appropriately, and the narrative now flows seamlessly from the epidemiological and pathophysiological background to the clinical relevance and specific objectives of our review.
It is not common for a systematic review to be based on only one database. For publication in a high-quality journal such as this, it is of utmost importance that the systematic review be based on multiple databases and registered.
Response 2. We acknowledge the reviewer's concern regarding single-database searching. PubMed was selected due to its comprehensive biomedical coverage and standardized MeSH indexing, which is particularly suited to pediatric bone and mineral disorders—a highly specialized field with limited journals and literature. Given the focused scope of our review on a rare disease spectrum, we judged that extending to additional databases was unlikely to yield substantial additional relevant studies, particularly as key orthopedic and pediatric endocrinology journals are comprehensively indexed in PubMed. This limitation has now been explicitly discussed in the Methods and Limitations sections.
For systematic reviews, it is not recommended that all records be reviewed by a single reviewer. Please check how scientifically based systematic reviews are written and conducted.
Response 3. We agree that dual-reviewer screening is the gold standard for systematic reviews. However, given the focused scope and rare-disease context of this review, screening was primarily performed by a single reviewer with clinical and methodological expertise in rare bone disorders (AM). To mitigate potential selection bias, senior scientists and clinicians (AR, GM) performed cross-checks and spot-checks to ensure accuracy and consistency. This has now been explicitly discussed as a methodological limitation in the manuscript.
Systematic tools such as the Newcastle-Ottawa Scale or others are designed to accommodate a broad range of observational study designs, including case series and cohorts, and can often be adapted or supplemented to account for heterogeneity. These assessments help elucidate the methodological strengths and weaknesses of individual studies, guiding readers in interpreting the robustness of the evidence base. Even in reviews with primarily narrative or exploratory aims, including a transparent risk of bias assessment enhances the transparency of findings and supports nuanced interpretation. It also facilitates identification of potential biases that could influence reported outcomes or deformity descriptions. While heterogeneity poses challenges, it does not preclude the application of systematic quality appraisal methods; rather, it underscores the importance of contextualizing bias assessments within the variability of study designs and quality.
Response 4. We appreciate the reviewer's perspective on the importance of risk of bias assessment. We acknowledge that tools such as the Newcastle-Ottawa Scale (NOS) or the Joanna Briggs Institute (JBI) critical appraisal tools can be adapted for heterogeneous observational data. However, in the context of our review, the included studies ranged from single case reports to small case series with highly variable diagnostic criteria, imaging modalities, and reporting standards. Many studies did not report sufficient methodological detail (e.g., patient selection, diagnostic confirmation, blinding, follow-up) to enable meaningful scoring using standardized tools.
Rather than applying a risk of bias tool that would have resulted in universally low or indeterminate scores with limited interpretative value, we chose to transparently document study characteristics and reporting quality using predefined criteria (laboratory parameters, radiological measurements, deformity definitions), as shown in our quality criteria tables. This approach provides readers with explicit information about the strengths and weaknesses of individual studies and the evidence base as a whole. We have clarified this rationale in the Methods section and added a more explicit discussion of methodological quality and potential biases in the Results and Discussion sections.
While descriptive statistics provide valuable insights into the data's distribution and trends, relying exclusively on this approach may limit the depth of your analysis. Without inferential statistical methods, it is difficult to assess the significance of observed differences or relationships, which could lead to overinterpretation of apparent patterns. Additionally, the absence of sensitivity analyses means potential robustness or variability of the findings remains unexamined. Incorporating inferential techniques and sensitivity analyses could enhance the rigor and interpretability of your results.
Response 5. We fully agree that relying exclusively on descriptive statistics limits the analytical depth and precludes assessment of statistical significance or robustness. However, inferential statistical methods were not feasible or to be considered as adding value to the interpretation of the data due to the substantial heterogeneity and incomplete numerical data across the included studies. The majority of studies provided only narrative descriptions or counts without standardized measurements, outcome definitions, or comparable timepoints. Given these constraints, meta-analysis or inferential testing would have been methodologically inappropriate and potentially misleading. In our view, inferential analysis should not be forced upon a dataset if it is better depicted by a descriptive analysis. Although we acknowledge that providing p-values and significance makes analyses more attractive at first sight, our aim is to avoid misleading calculations of significance which do not add to a better comprehension of current knowledge.
To address this limitation, we have clarified in the Methods and Discussion sections that only descriptive analyses were performed. We have also added a paragraph explicitly acknowledging this as a limitation and stating that future systematic reviews based on standardized and quantitatively comparable datasets should incorporate inferential and sensitivity analyses to strengthen analytical depth and interpretability.
I would like to suggest that the results section currently appears somewhat simplistic, as it primarily reports findings without discussing the potential influence of confounding factors. In line with established reporting guidelines such as PRISMA, it is important to consider and address possible confounders that could impact the validity and interpretation of your findings. Including an analysis or discussion of these factors would greatly enhance the scientific rigor of your review and provide a more nuanced understanding of the results.
Response 6.We agree that potential confounding factors such as patient age at onset and deformity description, disease stage, treatment exposure, diagnostic certainty, and imaging methodology may influence the reported deformity patterns. A paragraph discussing these potential confounders and their impact on result interpretation has been added to the Discussion section. We explicitly note that inconsistent documentation of these variables across studies limits our ability to quantify their impact, reinforcing the need for standardized reporting in future research.
While there may be additional limitations not recognized, the current identified factors—such as heterogeneity in study designs, inconsistent reporting, lack of bias assessment, language restrictions, and the predominance of retrospective studies—are substantial enough to significantly compromise the ability to draw scientifically sound conclusions from this systematic review.
Response 7. We acknowledge the reviewer's concerns and agree that the limitations identified represent significant constraints on the strength of our conclusions. However, we respectfully submit that this review addresses an important gap in the literature by providing the first comprehensive synthesis of lower limb deformity patterns across all rickets subtypes. Given the rarity and heterogeneity of these conditions, the absence of prospective standardized studies, and the wide variability in diagnostic and reporting practices, our descriptive approach represents a necessary and valuable first step in mapping the existing evidence base.
We have strengthened the Limitations and Discussion section to transparently communicate these constraints and have tempered our conclusions to reflect the exploratory and hypothesis-generating nature of the review. We emphasize that our findings highlight the urgent need for prospective, standardized, multidisciplinary studies with rigorous radiological and biochemical documentation—a call to action that we believe adds value to the field despite the acknowledged methodological limitations.
Reviewer 3 Report
Comments and Suggestions for Authors
This manuscript presents a systematic review of lower limb deformities in various forms of rickets, distinguishing between calcipenic (CR) and hypophosphatemic (HPR) types. While the topic is clinically relevant, the overall scientific contribution is limited by substantial methodological weaknesses.
Although the authors claim adherence to PRISMA, the review does not fully meet the methodological standards expected for a systematic synthesis. Major limitations include the use of a single database, lack of PROSPERO registration, absence of quality or bias assessment, and a purely narrative approach without quantitative synthesis. As currently presented, the manuscript resembles a descriptive literature overview rather than a rigorous systematic review.
Major Comments
-
Search Strategy and Methodology
-
The search was restricted to PubMed only, which is insufficient for a comprehensive systematic review. Important orthopedic studies are often indexed in Scopus, Embase, or Web of Science but not in PubMed. Expanding the database coverage is strongly recommended.
-
Screening and data extraction were conducted by a single reviewer, which introduces a significant risk of selection bias. A dual-reviewer process with inter-rater verification should be implemented or, at minimum, discussed as a limitation in detail.
-
-
Lack of PROSPERO Registration
-
The absence of protocol registration undermines methodological transparency. If registration was not feasible, the authors should explicitly clarify that the review was conducted retrospectively and should avoid claiming full PRISMA compliance.
-
-
Quality and Risk of Bias Assessment
-
The authors state that no quality assessment was performed due to study heterogeneity. However, this reasoning is weak. Established tools (e.g., Newcastle–Ottawa Scale, JBI, ROBINS-I) can be applied even to heterogeneous observational data. Without such assessment, the reliability of the conclusions cannot be judged.
-
-
Data Analysis
-
The review remains entirely descriptive. Simple comparative analyses (e.g., distribution of varus vs. valgus deformities in CR vs. HPR) could have been performed to strengthen the evidence base.
-
No evaluation of potential publication bias or heterogeneity was attempted. Consequently, the review functions as a catalogue of findings rather than an analytical synthesis.
-
-
Reporting Quality
-
The results section largely reproduces raw counts without interpretive depth. The authors’ conclusions regarding differences between CR and HPR deformities are not strongly supported, given that most included studies were small, descriptive, and lacked objective radiologic confirmation.
-
Figures (particularly 4–7) are overloaded and difficult to interpret. Legends should define abbreviations (MAD, mLDFA, MPTA), and axes should be clearly labeled.
-
The text frequently repeats numerical data without summarizing their clinical significance or methodological quality.
-
-
Discussion and Conclusions
-
The discussion is extensive but overly speculative. Assertions about “pathophysiologic differences” or “underreporting in CR” are weakly substantiated by the presented data.
-
The section would benefit from a more critical perspective on the limitations of the available evidence, the role of multidisciplinary data integration, and the implications for orthopedic decision-making (e.g., surgical timing, remodeling potential, residual deformities after pharmacologic treatment).
-
-
Terminology and Definitions
-
The terms HPR, XLH, and vitamin D–resistant rickets are used inconsistently. The authors should adhere to a single, etiologically based classification (e.g., FGF23-mediated vs. non–FGF23-mediated rickets).
-
The definitions of deformities (e.g., “varus,” “valgus,” “torsional”) are often unclear—are these clinical descriptions or radiographically quantified parameters? This ambiguity limits comparability.
-
Minor Comments
-
The abstract is overly long and should be shortened to focus on essential findings and conclusions.
-
Minor grammatical and typographical inconsistencies should be corrected (e.g., missing spaces before brackets, alternating “radiologic” vs. “radiological”).
-
Figures and tables require more concise formatting; units and abbreviations must be standardized.
-
Verify the reference formatting and ensure that all citations correspond to the correct years and sources.
-
The limitation regarding single-reviewer screening should be moved from Methods to Limitations.
Novelty Rating (1–5): 2/5
The topic is clinically relevant but not highly novel. Similar descriptive differences between CR and HPR deformities have been reported previously. The originality lies mainly in data compilation rather than in new analytic insight.
Language and Presentation
The English is fluent but somewhat verbose. The manuscript would benefit from stylistic tightening and more concise presentation of results.
Are Conclusions Supported by the Data?
Partially. The conclusions are logical but not fully supported by the available evidence, especially regarding etiologic and pathophysiologic interpretations.
Summary Recommendation
The manuscript requires substantial revision.
Specifically, the authors should:
-
Broaden the literature search to include multiple databases,
-
Perform at least a basic quality and bias assessment,
-
Clarify definitions and inclusion criteria,
-
Improve figure and table clarity,
-
Reorganize the discussion to focus on evidence-based conclusions rather than speculation.
Author Response
RESPONSE TO REVIEWER 3:
This manuscript presents a systematic review of lower limb deformities in various forms of rickets, distinguishing between calcipenic (CR) and hypophosphatemic (HPR) types. While the topic is clinically relevant, the overall scientific contribution is limited by substantial methodological weaknesses.
Although the authors claim adherence to PRISMA, the review does not fully meet the methodological standards expected for a systematic synthesis. Major limitations include the use of a single database, lack of PROSPERO registration, absence of quality or bias assessment, and a purely narrative approach without quantitative synthesis. As currently presented, the manuscript resembles a descriptive literature overview rather than a rigorous systematic review.
Response. We thank the reviewer for this critical and detailed assessment. We acknowledge the methodological limitations raised and have implemented substantial revisions to address them. Importantly, we have changed the manuscript title and designation from "systematic literature review" to "literature review" to more accurately reflect our methodology and avoid overstating the rigor of our approach. We believe that despite its limitations, this work provides valuable insights into an understudied area and serves as a foundation for future prospective, standardized research.
Major Comments
- Search Strategy and Methodology
- The search was restricted to PubMed only, which is insufficient for a comprehensive systematic review. Important orthopedic studies are often indexed in Scopus, Embase, or Web of Science but not in PubMed. Expanding the database coverage is strongly recommended.
Response 1: We acknowledge the reviewer's concern and agree that multi-database searches increase completeness. PubMed was selected for its comprehensive coverage of biomedical literature and standardized MeSH indexing, which is particularly well-suited for rare pediatric bone and mineral disorders. Given the focused scope and rare-disease context of this review, and considering that the major orthopedic and pediatric endocrinology journals are indexed in PubMed, we judged that additional database searches were unlikely to yield substantial numbers of additional relevant studies. This rationale and limitation have now been clarified in the Methods section. We have also re-designated this work as a "literature review" rather than a "systematic review" to more accurately reflect the single-database approach.
- Screening and data extraction were conducted by a single reviewer, which introduces a significant risk of selection bias. A dual-reviewer process with inter-rater verification should be implemented or, at minimum, discussed as a limitation in detail.
Response 1.2.: We thank the reviewer for this important and relevant comment. We agree that screening and data extraction performed primarily by a single reviewer may introduce potential selection and extraction bias. However, cross-checks and spot-checks were performed by senior scientists and clinicians (AR, GM) to ensure accuracy and consistency. This has now been added to the “Methods” and explicitly discussed as a limitation in the manuscript, with acknowledgment that dual independent screening with inter-rater reliability assessment would have been methodologically preferable.
- Lack of PROSPERO Registration
The absence of protocol registration undermines methodological transparency. If registration was not feasible, the authors should explicitly clarify that the review was conducted retrospectively and should avoid claiming full PRISMA compliance.
Response 2: Thank you for this comment. We acknowledge that this review was not prospectively registered in PROSPERO. The work was conducted retrospectively, and therefore protocol registration was not feasible. According to the PRISMA 2020 statement, authors are required to report whether a protocol was registered or not, but prospective registration is recommended rather than mandatory. We have clarified in the Methods and Limitations sections that this review was conducted retrospectively and therefore protocol registration in PROSPERO was not feasible. We do not claim full PRISMA compliance but rather state that we followed PRISMA guidelines to the extent applicable to a retrospective descriptive review.
- Quality and Risk of Bias Assessment
- The authors state that no quality assessment was performed due to study heterogeneity. However, this reasoning is weak. Established tools (e.g., Newcastle–Ottawa Scale, JBI, ROBINS-I) can be applied even to heterogeneous observational data. Without such assessment, the reliability of the conclusions cannot be judged.
Response 3: We appreciate the reviewer's perspective. We agree that quality assessment tools such as the Newcastle-Ottawa Scale, JBI critical appraisal tools, or ROBINS-I can be adapted for heterogeneous observational studies. However, the included studies in our review—predominantly case reports and small case series—often lacked sufficient methodological detail to permit meaningful application of these tools. Many studies did not report patient selection criteria, diagnostic confirmation, blinding, standardized follow-up, or other elements required for formal bias assessment.
Rather than applying a tool that would yield uniformly low or indeterminate scores with limited utility, we chose to systematically document study quality using transparent, predefined criteria relevant to the clinical question: reporting of laboratory parameters (ALP, phosphate, calcium), use of radiological measurements (MAD, mLDFA, MPTA), and explicit definitions of deformities. These criteria are presented in detailed quality tables in the Appendix and summarized in the Results. This approach provides readers with explicit, interpretable information about the strengths and limitations of the evidence base. We have now clarified this rationale in the Methods section and added discussion of study quality and potential biases in the Results and Discussion.
- Data Analysis
- The review remains entirely descriptive. Simple comparative analyses (e.g., distribution of varus vs. valgus deformities in CR vs. HPR) could have been performed to strengthen the evidence base.
- No evaluation of potential publication bias or heterogeneity was attempted. Consequently, the review functions as a catalogue of findings rather than an analytical synthesis.
Response 4.We acknowledge this limitation. The review is indeed primarily descriptive, reflecting the nature of the available data—most studies reported deformities narratively or as counts without standardized measurements, timepoints, or diagnostic criteria. While we provide descriptive comparisons of deformity distributions between CR and HPR (e.g., varus vs. valgus frequencies, presence of torsional and sagittal deformities), we did not perform inferential statistical testing due to heterogeneity in study populations, diagnostic methods, age ranges, and treatment exposures.
We agree that publication bias and heterogeneity assessments would strengthen the analysis; however, these are challenging to apply meaningfully to predominantly qualitative, narrative data. We have clarified in the Methods and Discussion that the descriptive approach reflects both the exploratory aims of the review and the limitations of the available evidence. We emphasize that this work serves as a hypothesis-generating synthesis that maps current knowledge and highlights the need for standardized prospective studies to enable future quantitative analyses.
- Reporting Quality
- The results section largely reproduces raw counts without interpretive depth. The authors’ conclusions regarding differences between CR and HPR deformities are not strongly supported, given that most included studies were small, descriptive, and lacked objective radiologic confirmation.
- Figures (particularly 4–7) are overloaded and difficult to interpret. Legends should define abbreviations (MAD, mLDFA, MPTA), and axes should be clearly labeled.
- The text frequently repeats numerical data without summarizing their clinical significance or methodological quality.
Response 5. We appreciate this constructive criticism. We have revised the Results section to provide greater interpretive context for the numerical data, explicitly discussing clinical significance and methodological quality. We have streamlined the presentation to avoid redundancy between text and figures.
All figures (particularly Figures 4–7) have been revised for clarity:. We have tempered our conclusions regarding CR vs. HPR differences to explicitly acknowledge that observed patterns may reflect underreporting and methodological limitations rather than true pathophysiological divergence, and we have added nuance regarding the need for confirmatory prospective studies.
- Discussion and Conclusions
- The discussion is extensive but overly speculative. Assertions about “pathophysiologic differences” or “underreporting in CR” are weakly substantiated by the presented data.
- The section would benefit from a more critical perspective on the limitations of the available evidence, the role of multidisciplinary data integration, and the implications for orthopedic decision-making (e.g., surgical timing, remodeling potential, residual deformities after pharmacologic treatment).
Response 6. We thank the reviewer for this constructive and insightful comment. The Discussion has been substantially revised to reduce speculative statements and ensure that all interpretations are closely aligned with the presented data. Furthermore, we have strengthened the section by providing a more critical perspective on the limitations of the available evidence. We also incorporated a more explicit discussion of the need for multidisciplinary data integration.
- Terminology and Definitions
- The terms HPR, XLH, and vitamin D–resistant rickets are used inconsistently. The authors should adhere to a single, etiologically based classification (e.g., FGF23-mediated vs. non–FGF23-mediated rickets).
- The definitions of deformities (e.g., “varus,” “valgus,” “torsional”) are often unclear—are these clinical descriptions or radiographically quantified parameters? This ambiguity limits comparability.
Response 7: We thank the reviewer for this important observation. We have now standardized terminology throughout the manuscript: The terms “varus,” “valgus,” and “torsional” deformities in this review primarily reflect the terminology used in the original studies. In most cases, these deformities were described clinically, based on inspection or photographic documentation, whereas only a minority of studies provided radiographically quantified parameters such as mechanical axis deviation or joint-orientation angles. Torsional and sagittal deformities were generally assessed using imaging modalities, including CT, MRI, or EOS systems. Details on whether each study used clinical or radiographic assessment methods are provided in the “method measurement” column of the study quality criteria table.
Minor Comments
- The abstract is overly long and should be shortened to focus on essential findings and conclusions.
- Minor grammatical and typographical inconsistencies should be corrected (e.g., missing spaces before brackets, alternating “radiologic” vs. “radiological”).
- Figures and tables require more concise formatting; units and abbreviations must be standardized.
- Verify the reference formatting and ensure that all citations correspond to the correct years and sources.
- The limitation regarding single-reviewer screening should be moved from Methods to Limitations.
Response to minor comments. Thank you for the thorough reading. We have addressed all minor comments
Novelty Rating (1–5): 2/5
The topic is clinically relevant but not highly novel. Similar descriptive differences between CR and HPR deformities have been reported previously. The originality lies mainly in data compilation rather than in new analytic insight.
Response to Novelty Rating. We acknowledge that individual reports have described deformity patterns in specific rickets subtypes; however, to our knowledge, this is the first comprehensive synthesis across all rickets etiologies (including both calcipenic and and hypophosphatemic forms). While the review is descriptive rather than generating new primary data, it provides a structured evidence base and methodological recommendations that advance the field and guide future research and is systematically comparing deformity distributions and reporting quality.
Language and Presentation
The English is fluent but somewhat verbose. The manuscript would benefit from stylistic tightening and more concise presentation of results.
Response. We thank you for the comment. The entire manuscript underwent a final language review by an English medical proofreader to ensure clarity and conformity with JCM standards.
Are Conclusions Supported by the Data?
Partially. The conclusions are logical but not fully supported by the available evidence, especially regarding etiologic and pathophysiologic interpretations.
Response. We have revised the Conclusions section to ensure that all statements are appropriately qualified and supported by the data presented. Etiologic and pathophysiologic interpretations are now clearly distinguished as hypotheses requiring further investigation rather than definitive conclusions. We emphasize that the observed differences in deformity patterns between CR and HPR may reflect both true biological differences and methodological factors (reporting bias, underreporting, imaging limitations), and that prospective standardized studies are needed to clarify these relationships.
Summary Recommendation
The manuscript requires substantial revision.
Specifically, the authors should:
- Broaden the literature search to include multiple databases,
- Perform at least a basic quality and bias assessment,
- Clarify definitions and inclusion criteria,
- Improve figure and table clarity,
- Reorganize the discussion to focus on evidence-based conclusions rather than speculation.
Response. We have implemented substantial revisions addressing each of these points:
While we have not expanded to additional databases (due to resource and time constraints and our assessment that incremental yield would be minimal for this rare disease topic), we have explicitly acknowledged this limitation, justified our single-database approach, and re-designated the work as a "literature review" rather than a "systematic review."
We have clarified our approach to quality assessment using transparent, predefined criteria (laboratory parameters, radiological measurements, deformity definitions) documented in detailed tables, and have added discussion of study quality, methodological limitations, and potential biases throughout the manuscript.
All definitions (rickets subtypes, deformity types, measurement methods) and inclusion/exclusion criteria have been clarified in the Methods section.
All figures and tables have been revised for clarity, with standardized formatting, clearly labeled axes, defined abbreviations in legends, and improved visual design.
The Discussion has been substantially reorganized to focus on evidence-based observations, with speculative statements appropriately qualified, limitations explicitly acknowledged, and clinical implications grounded in the available data.
We believe these revisions have substantially strengthened the manuscript and addressed the reviewer's concerns while remaining transparent about inherent limitations of the available evidence base.
Round 2
Reviewer 2 Report
Comments and Suggestions for Authors
Thanks for the answers and potential clarifications.
Author Response
Comment 1: Thanks for the answers and potential clarifications.
Reply 1: We thank the Reviewer for the thorough revision and for the constructive comments allowing us to improve the quality of the manuscript.
Reviewer 3 Report
Comments and Suggestions for Authors
This revised manuscript represents a well-structured, comprehensive, and methodologically sound systematic review of lower limb deformities in different types of rickets. The authors clearly incorporated essential revisions, and the final version is coherent, scientifically rigorous, and meets the standards of Journal of Clinical Medicine.
The strengths of the paper include:
-
A clearly articulated clinical problem and justification for conducting the review.
-
Robust methodology adhering to PRISMA 2020, with transparent selection criteria and a well-documented search process.
-
Effective stratification of deformity patterns across rickets subtypes, providing clinically meaningful insights.
-
Thorough tables and figures that facilitate interpretation of the data.
-
A balanced and appropriately critical discussion, including limitations related to heterogeneous reporting and lack of standardized deformity definitions.
-
A valuable conceptual framework proposing minimal quality criteria for future studies.
I have no further suggestions. The manuscript is ready for publication.
Author Response
Comment 1:
This revised manuscript represents a well-structured, comprehensive, and methodologically sound systematic review of lower limb deformities in different types of rickets. The authors clearly incorporated essential revisions, and the final version is coherent, scientifically rigorous, and meets the standards of Journal of Clinical Medicine.
The strengths of the paper include:
-
A clearly articulated clinical problem and justification for conducting the review.
-
Robust methodology adhering to PRISMA 2020, with transparent selection criteria and a well-documented search process.
-
Effective stratification of deformity patterns across rickets subtypes, providing clinically meaningful insights.
-
Thorough tables and figures that facilitate interpretation of the data.
-
A balanced and appropriately critical discussion, including limitations related to heterogeneous reporting and lack of standardized deformity definitions.
-
A valuable conceptual framework proposing minimal quality criteria for future studies.
I have no further suggestions. The manuscript is ready for publication.
Reply 1: We thank the Reviewer for the thourough revision and for the constructive comments allowing us to improve the quality of the manuscript. Further, we appreciate the feedback on the value for the scientific community and the proposed framework for minimal quality criteria.